# Demonstration of an AI-driven workflow for autonomous high-resolution scanning microscopy

Saugat Kandel [1] ✉, Tao Zhou[2], Anakha V. Babu[3], Zichao Di[4], Xinxin Li [2,5], Xuedan Ma [2,5], Martin Holt [2], Antonino Miceli [1], Charudatta Phatak [6] & Mathew J. Cherukara [1] ✉

Modern scanning microscopes can image materials with up to sub-atomic spatial and sub-picosecond time resolutions, but these capabilities come with large volumes of data, which can be difficult to store and analyze. We report the Fast Autonomous Scanning Toolkit (FAST) that addresses this challenge by combining a neural network, route optimization, and efficient hardware controls to enable a self-driving experiment that actively identifies and measures a sparse but representative data subset in lieu of the full dataset. FAST requires no prior information about the sample, is computationally efficient, and uses generic hardware controls with minimal experiment-specific wrapping. We test FAST in simulations and a dark-field X-ray microscopy experiment of a $WSe_2$ film. Our studies show that a FAST scan of <25% is sufficient to accurately image and analyze the sample. FAST is easy to adapt for any scanning microscope; its broad adoption will empower general multi-level studies of materials evolution with respect to time, temperature, or other parameters.

Scanning microscopes are versatile instruments that use photons, electrons, ions, neutrons, or mechanical probes to interrogate atomic-scale composition, topography, and functionality of materials, with up to sub-atomic spatial resolution and sub-picosecond time resolution[1–3]. Notwithstanding the variation in the probe modalities, these instruments all rely on a scan of the sample to generate spatially resolved signals that are then collected to form an image of the sample. Ongoing advances in instrumentation, such as the development of next-generation X-ray and electron detectors[4,5], have meant that scanning microscopes can now image faster and at higher resolutions than ever before. We can now envision a broad use of these instruments to study not only static systems but also multi-level studies of the dynamic evolution of materials with time, temperature, or other parameters, even in situ or operando[6]. Fine-resolution large-field-of-view scanning experiments, however, come with some significant drawbacks: the volume of data generated and the probe-induced damage to the sample can be prohibitively large. For example, it is now routinely possible to perform X-ray imaging of 1 mm³ volumes at ≈10 nm resolution, but this generates ≈$10^{15}$ voxels of data[7,8] and requires a commensurately high probe dose[9]. Meanwhile, the information of interest in these experiments is often concentrated in sparse regions that contain interfaces, defects, or other specific structural elements. Directing the scan to only these locations could greatly reduce the scan time and data volume, but it is difficult to obtain this information a priori. Addressing this challenge with a human-in-the-loop protocol, where an experienced user examines the data acquired to identify trends and guide the scan, can be tedious and prohibitively time-consuming (in comparison to the experimental acquisition time). Given these factors, the development of autonomous acquisition techniques that can continuously analyze acquired data and drive the sampling specifically toward regions of interest is imperative so as to make full use of the potential of these scientific instruments.

[1]Advanced Photon Source, Argonne National Laboratory, Lemont, IL 60439, USA. [2]Nanoscience and Technology Division, Argonne National Laboratory, Lemont, IL 60439, USA. [3]KLA Corporation, Ann Arbor, MI 48105, USA. [4]Mathematics and Computer Science, Argonne National Laboratory, Lemont, IL 60439, USA. [5]Consortium for Advanced Science and Engineering, University of Chicago, Chicago, Illinois 60637, USA. [6]Materials Science Division, Argonne National Laboratory, Lemont, IL 60439, USA. ✉e-mail: skandel@anl.gov; mcherukara@anl.gov

In parallel to the advances in scientific instrumentation, the last decade has also seen the rapid development of deep learning (DL) techniques and their applications in all domains of science and technology, including for the acceleration and enhancement of advanced microscopy methods[10–13]. These DL-based inversion methods are enabling real-time data analysis, which is, in turn, opening the door to self-driving techniques that make real-time acquisition decisions based on real-time data streams. Such self-driving or autonomous experimentation methods[14] are methods that combine automated experimental control with on-the-fly data-driven decision-making so that an algorithm adaptively explores parameter spaces of interest and conducts new experiments until it achieves a pre-defined completion criterion[15]. These methods therefore have the potential to not only remove the need for constant human supervision and intervention in experiments but also make optimal choices in parameter spaces that are too large for humans to easily contextualize. As such, they have the potential to revolutionize experimental design in many scientific fields, including the field of imaging and materials characterization.

In general, the use of data-driven priors to direct future experiments is a Bayesian search problem, for which the use of off-the-shelf deep learning methods usually do not suffice[16]. Specific to microscopy, a popular Bayesian search approach is to use unsupervised (without pre-training) Gaussian Processes (GPs) that could continuously determine the spatial locations that we are most uncertain about, then direct the scanning to these locations[17–22]. While GPs are powerful techniques, their computational cost tends to scale cubically with the number of points acquired. The decision-making time increases during the experiment and quickly exceeds the acquisition time for the measurement itself. The development of scalable GPs is a significant area of research, but these methods are not yet ready for application in large-scale imaging problems[23]. General supervised alternatives such as reinforcement learning can be powerful and fast, but they often require costly pre-training and tend to ignore the global state of the

parameter space in exchange for a local search; as such, they have only found limited traction for scanning imaging modalities[24].

Specifically for scanning microscopy applications, Godaliyadda et al.[25] have proposed to achieve computationally efficient autonomous sampling with the Supervised Learning Approach for Dynamic Sampling (SLADS) technique. The SLADS technique uses curated feature maps to quantify the current measurement state and predict the total image quality improvement obtained by measuring a given point, thereby informing the choice of which point to measure next. Variations of this technique have found applications in live steering for dose-efficient crystal positioning for crystallography[26] and for imaging with transmission electron microscopy[27] and mass spectrometry[28] methods. These works, however, either involve training with and reconstruction of binary images only[26,27] or require extensive training with images closely related to the sample under study[28]. As such, they are difficult to translate to imaging settings with more complex images, particularly for imaging without any prior assumptions about the sample. Meanwhile, Zhang et al.[29] have incorporated a neural network (NN) within the SLADS method (for the SLADS-Net method) and shown in numerical experiments that it is sufficient to train the method on only a generic image, eschewing any prior knowledge about the sample, to produce a high-fidelity image with sparse sampling. However, this has not yet been demonstrated in experiment.

In this work, we report the **F**ast **A**utonomous **S**canning **T**oolkit (FAST) that combines the SLADS-Net method, a route optimization technique, and efficient and modular hardware controls to make on-the-fly sampling and scan path choices for synchrotron-based scanning microscopy. This method relies on sample-agnostic training to dynamically measure and reconstruct a complicated (non-binary) sample, distinguishing this toolkit from existing SLADS-based workflows. Moreover, its computational cost is negligible compared to the acquisition time, even when run on a low-power edge computing device placed at a synchrotron beamline, which presents a significant advantage over more generic autonomous experimentation techniques. These characteristics enable the application of our workflow in the high-precision nanoscale scanning X-ray microscopy instrument present at the hard X-ray nanoprobe beamline at the Advanced Photon Source.

We validate the FAST scheme through real-time demonstration at the hard X-ray nanoprobe beamline at the APS[30]. A few-layer exfoliated two-dimensional $WSe_2$ thin film was chosen as a representative example; the preparation process for the thin film often leaves microscopic air bubbles trapped underneath the thin film, deforming the 2D material. We show that an adaptive scan of <25% of the sample is sufficient to produce a high-fidelity reconstruction that identifies all the bubbles within the field of view and even to acquire quantitative information about the film curvature induced by these bubbles. The scheme quickly identifies the deformed part of the 2D material and focuses its attention there while ignoring regions of the film that are flat and homogeneous. Film curvature reconstructed from the adaptive scan (<25% coverage) is consistent with that reconstructed from the full-grid scan (100% coverage). Given these characteristics, the FAST scheme can be directly applied in other scanning techniques and instruments at the APS and elsewhere and may underpin the development of many multi-level experimental studies.

## Results

Figure 1 shows the experimental setup that scans a focused X-ray beam on a sample while acquiring a two-dimensional diffraction image at each point. The live demonstration was performed on a few-layer $WSe_2$ sample with the detector placed along the 008 Bragg peak, with $2\theta = 43.1°$ at 10.4 keV. The diffraction patterns were processed on the detector computer (see "Methods") to generate the integrated intensities for use in the FAST workflow. The final output of the workflow is a dark-field image of the $WSe_2$ sample.

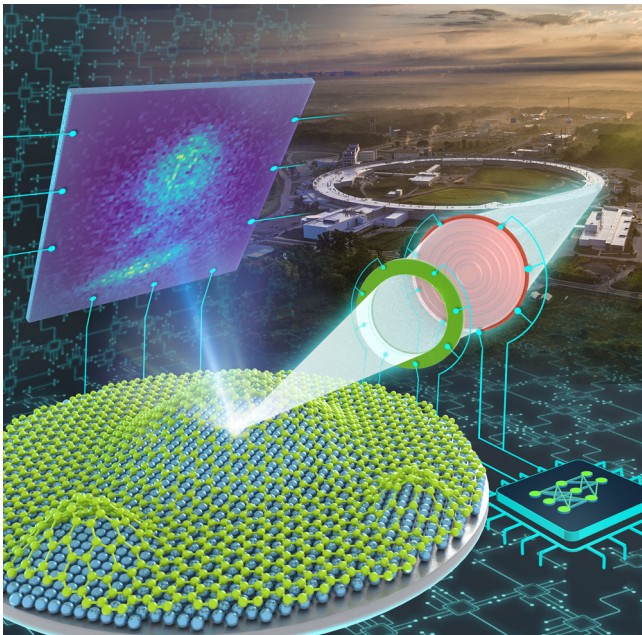

**Fig. 1 | Artist's representation of the autonomous dark-field scanning microscopy experiment at the Advanced Photon Source (APS).** The APS synchrotron produces a coherent X-ray beam that is focused using a zone plate setup. It strikes a $WSe_2$ film (green) exfoliated onto a Si substrate (blue), which generates diffraction patterns that are collected by a two-dimensional detector. Above the bubbles, the lattice of the film rotates, shifting the diffracted intensities away from its nominal positions. The beam position, as well as the detector acquisition, are autonomously controlled by the FAST AI-based workflow. Image by Argonne National Laboratory.

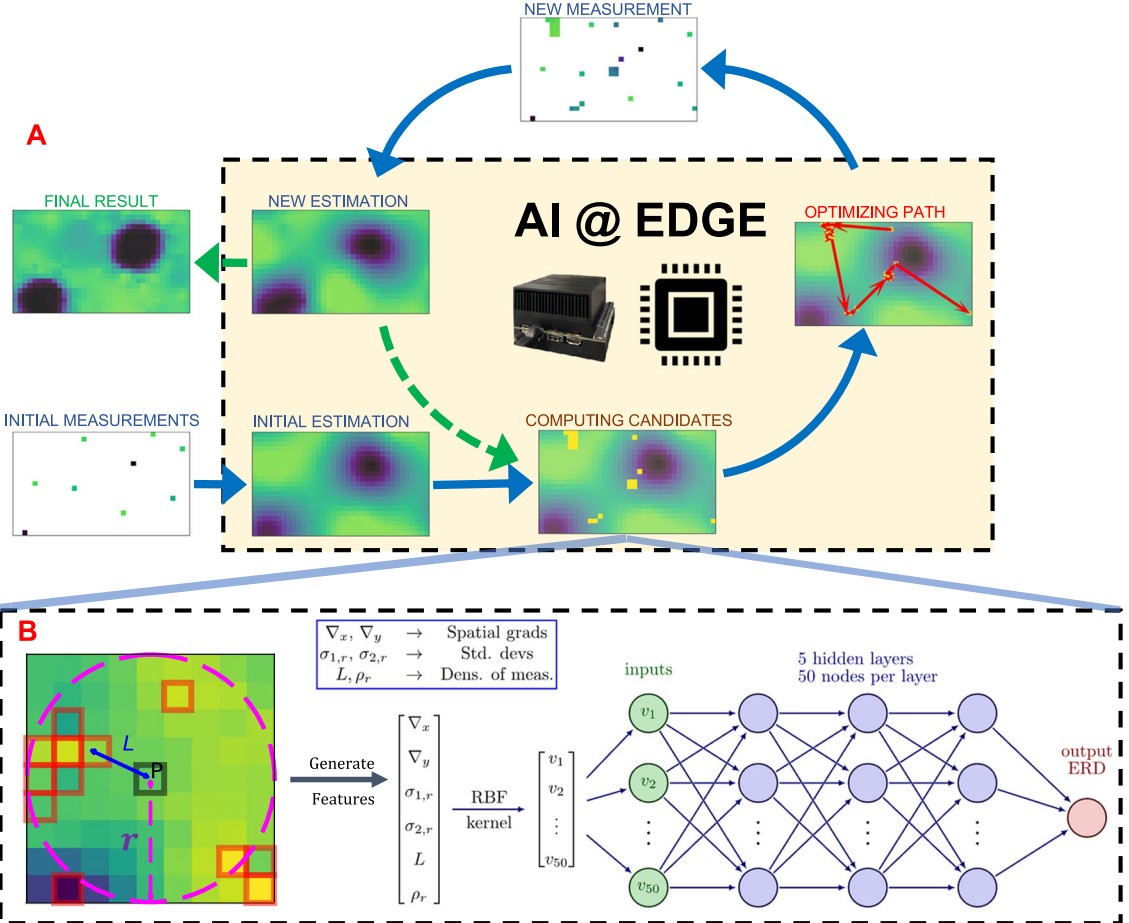

**Fig. 2 | The FAST workflow. A** A set of quasi-random initial measurements are transferred to the edge device, which sequentially generates an initial sample estimate, computes the candidate points to be measured next, and calculates the travel path for the measurement. The new measurements are combined with the existing measurements and used to calculate a new estimate, and the process is repeated until it achieves a completion criterion. **B** The candidate computation starts by examining the local neighborhood (with radius $r$) of each unmeasured point $P$, with the highlighted points indicating points already measured, to generate a 6-dimensional feature vector. The feature vector is transformed to a 50-dimensional vector using the Radial Basis Function (RBF) kernel and used as input to a multi-layer NN. The NN then predicts the expected improvement in the image (ERD) from measuring the point $P$. A set of unmeasured pixels with the highest ERD are selected as candidates for the next measurement.

## Self-driving scanning microscopy workflow

Figure 2A broadly illustrates the FAST workflow for the experiments reported here. To initiate the workflow, a low-discrepancy quasi-random selection (generated using the Hammersely sequence[31]) of sample position is measured corresponding to 1% of the total area of interest. The integrated intensities of the measurements are transferred to the edge device, an NVIDIA Jetson Xavier AGX located adjacent to the detector, which uses Inverse Distance Weighted (IDW) interpolation to estimate the dark-field image. The estimated image serves as input for the decision-making step whereby the prospective measurement points are identified.

This self-driving workflow adopts the Supervised Learning Approach for Dynamic Sampling using Deep Neural Networks (SLADS-Net) algorithm[29] to find the prospective measurement points. In effect, the SLADS-Net algorithm uses the current measurements to identify the best unmeasured points that, when added to the existing dataset, would have the greatest effect on the quality of the reconstructed image. As illustrated in Fig. 2B, this is accomplished by, first, representing each unmeasured point as a feature vector with elements that depend on the measurement state in the neighborhood of the point. These feature vectors are used as input for a pre-trained neural network with 5 hidden layers, with 50 nodes per layer, and with the ReLU activation function. The neural network then predicts the expected reduction in distortion (ERD), a metric (loosely speaking) for the

expected improvement in the reconstruction quality obtained from measuring this unmeasured point, individually for each unmeasured point. The original SLADS-Net algorithm simply uses the unmeasured point with the highest ERD for the next measurement and repeats this procedure pointwise. In practice, if the measurement procedure and the motor movements are fast, then the ERD calculation also has to be commensurately fast to reduce the dead-time in the experiment. In this work, we mitigate this requirement by instead selecting a batch of points that have the highest ERD, sorted in descending order—we found that a batch of 50 points adequately minimized the experimental dead-time while still ensuring that the overall measurement was adequately sparse.

The coordinates of these 50 points are passed on to a route optimization algorithm based on Google's OR-Tools[32] to generate the shortest path for the motors to visit all of them. This path is appended to the look-up table in the EPICS[33] scan record, which then kicks off the data acquisition. Henceforth, the scan is automatically paused after every 50 points, raising a flag that triggers a callback function on the edge device. There, a new estimated dark field image of the sample is generated, and the coordinates for the next 50 prospective points are computed. The scan is resumed after the EPICS scan record receives the new coordinates for the optimized scanning path. The actual scanning of the focused X-ray beam is achieved by moving two piezoelectric linear translation motors in step mode.

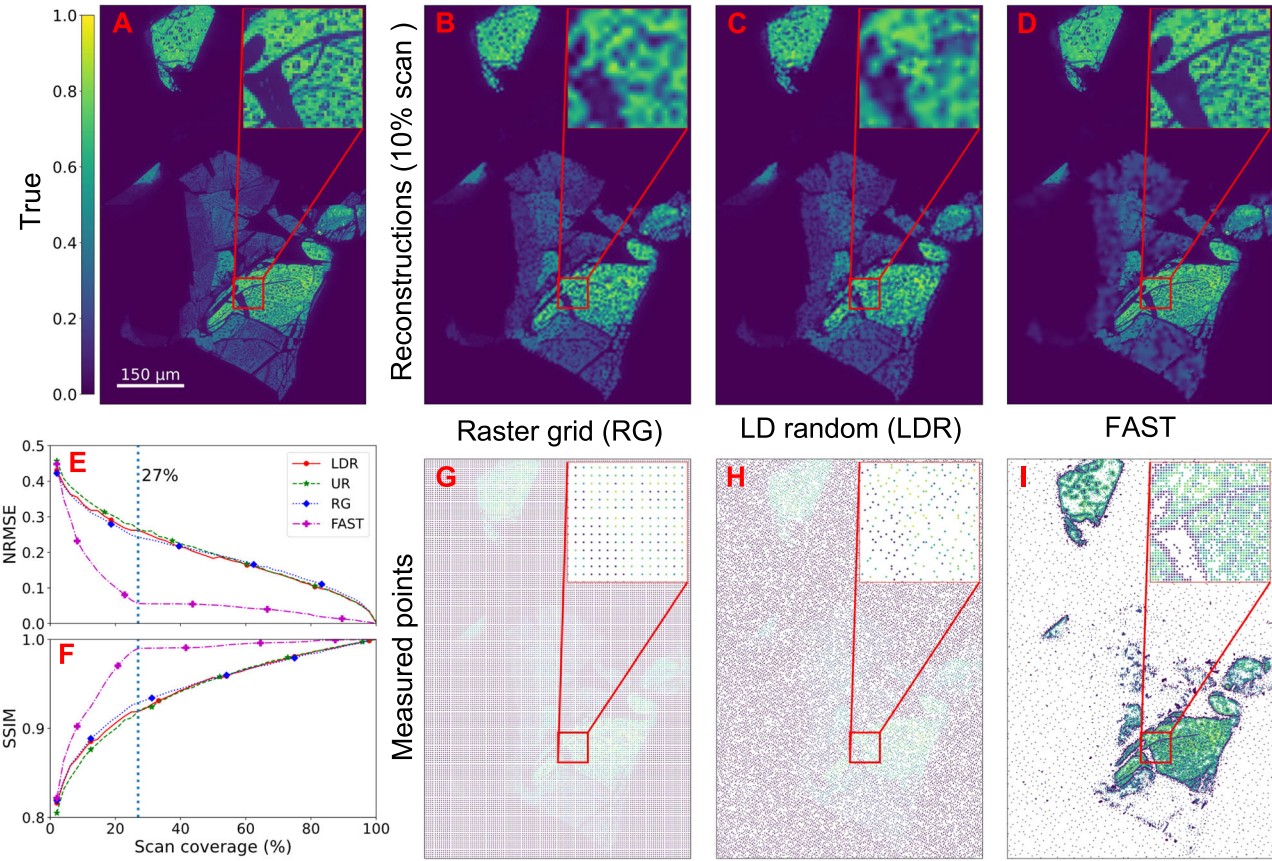

**Fig. 3 | Numerical comparison of sampling methods.** **A** shows the ground truth with the color scale representing the normalized intensity, **B**–**D** show respectively the raster grid (RG), low-discrepancy random (LDR), and FAST reconstructions at 10% scan coverage, and **G**–**I** show the actual scan points that produce these reconstructions. **E**, **F** show the evolution of the normalized root mean square error (NRMSE), for which lower is better, and the Structural Similarity metric (SSIM), for which higher is better, as a function of the scan coverage. The FAST reconstruction stabilizes at 27% coverage, while the other techniques take significantly longer to reach the same quality. Source data are provided as a Source Data file.

The detector exposure time is set to 0.5 s and comes with an overhead of 0.2 s.

For the 200 × 40 pixels object described in "Results: Experimental demonstration", the workflow required ≈0.15 s to compute the new positions, ≈42 s to scan the set of 50 positions, and a total of ≈0.37 s to process the diffraction patterns and communicate the measurements. This represents an overhead of ≲2%. The workflow is currently entirely CPU-bound, relying on the on-board 8-core ARM CPUs, and does not take advantage of the GPU bundled into the NVIDIA AGX device. These timing results showcase the rapid data-driven decision-making ability that is characteristic of the FAST workflow. In the future, we expect to perform the computation in a parallelized and asynchronous fashion, which would further reduce this overhead.

We also note that, for all the results reported in this work, the underlying NN was trained on the standard "cameraman"[34] image that has no relation to microscopy, and we discuss the choice of a training image in Supplementary Information S.2. For details about the SLADS-Net algorithm and the sample-agnostic training procedure, the reader is referred to the "Methods" section.

**Numerical demonstration for scanning dark-field microscopy**

We first validated the performance of the proposed workflow through a numerical experiment on a set of pre-acquired dark-field microscopy data. Here, we compared the FAST sampling with three static sampling techniques:

1. **Raster grid (RG):** For a test sampling percentage, we generated an equally spaced raster grid that provides a uniform coverage of the sample.

2. **Uniform random (UR) sampling:** The measurement pixels were drawn from a uniform random distribution.

3. **Low-discrepancy (LDR) quasi-random sampling:** For each measurement percentage, we generated a low-discrepancy sampling grid using the quasi-random Hammersly sequence.

The test dataset is a dark field image of size 600 × 400 pixels which represents 240,000 possible measurement positions. This covers a physical area of 900 × 600 μm and encloses multiple flakes of $WSe_2$ with various thicknesses, with the thicker regions associated with regions of higher brightness in the image (Fig. 3). At this spatial resolution, only medium and large-sized bubbles (with diameter >2 μm) can be observed. As explained previously, the bubbles deform the surface and shift the Bragg peak of the 2D materials away from their theoretical (flat region) positions, resulting in regions of darker contrast. Finally, the image also contains flake-free regions that have zero integrated intensities.

For this comparison, we first initialized the FAST sampling with a 1% measurement coverage (as described above), then successively measured 50 additional points at every iteration. For each FAST measurement, we also generate RG, UR, and LDR measurement masks with the same number of scan points. In this fashion, we generate a sequence of sampling masks and the associated reconstructions until we achieve 100% sampling.

We present the numerical results in Fig. 3, where we show a comparison of the various methods at 10% sampling. Note that while the proposed method internally uses the fast IDW algorithm for the inpainting, the final images presented here are calculated using the

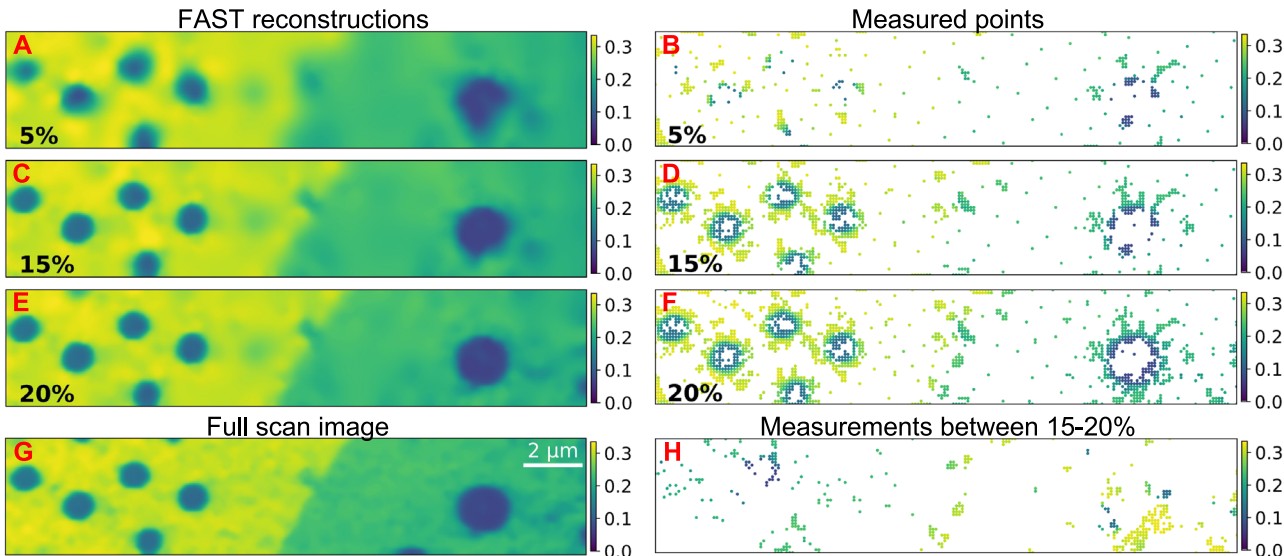

**Fig. 4 | Evolution of the FAST scan. A**, **C**, **E** show the reconstruction at 5%, 15%, and 20% reconstructions, respectively, **B**, **D**, **F** show the corresponding actual measurement points. **G** shows the image obtained through a full-grid pointwise scan. The color scale in (**A**–**G**) shows the normalized intensities. **H** shows only the points sampled between 15 and 20% coverage.

higher-quality biharmonic inpainting technique[35]. The uniform random scheme performs worse than the LDR and raster grid schemes and is not shown in the figure. In Fig. 3A–D, we can see that the FAST sampling is able to reproduce with high fidelity the flake boundaries, the bubbles, and the regions of transition between the varying levels of thicknesses. In contrast, the LDR and raster schemes produce much lower-quality reconstructions of these features. Figure 3E shows an evolution of the normalized root mean squared error (NRMSE), and Fig. 3F the structural similarity metric (SSIM) (which measures multiscale perceptual similarity) for the different sampling techniques. It is evident that FAST produces high-quality reconstructions at much lower measurement percentages than the examined static sampling techniques. We note that the result could be further improved in the future by using a more sophisticated inpainting technique within the FAST method. To understand how FAST outperforms the other methods under the same sampling condition, we show the actual measured positions of the various schemes at 10% coverage (Fig. 3G–I). FAST preferentially samples the regions with significant heterogeneity over the homogeneous regions. This is particularly useful for sparse samples, where the time spent sampling from empty regions adds little additional information.

**Experimental demonstration**

We next demonstrate the application of the FAST workflow in a live experiment at a synchrotron beamline. A video showing the sampling, recorded live during the actual experiment, is available here[36]. Other than starting the workflow scripts at the beginning, the entire experiment was unmanned and fully automated. In order to measure the deformed $WSe_2$ flakes in detail, a higher spatial resolution of 100 nm was chosen. This limits the field of view to $20 \times 4$ μm for a scan point density of $200 \times 40$ points.

In Fig. 4, we show the reconstructed dark field image (subplots A, C, E) and the measurement points (subplots B, D, F) from 5 to 20% coverage and compare them to that obtained from raster scanning the sample with 100% coverage (subplot G). We see that the FAST method identifies some of the regions of heterogeneity—the edges of the bubbles—and starts to preferentially sample these regions within 5% coverage of the sample. At 15% coverage, these regions are extensively sampled. The reconstruction does not change significantly between

15 and 20%, indicating that the reconstruction has stabilized. Moreover, the 20% reconstruction also contains sharp and accurate reproductions of all the major features present in the full scan image.

A point of interest is that the partially scanned bubble at the bottom right corners of Fig. 4E–G shows up only in the 20% scan and not in the 15% scan. To explain this, we note that the 5% scan, and therefore the initial 1% quasi-random sampling, does not contain any measurements in the neighborhood of this bubble. The FAST scheme favors the exploitation of regions it knows to be heterogeneous over the exploration of this fully unknown region and therefore only explores this region much later in the measurement process (Fig. 4H). This is, in fact, an instance of the general exploration-exploitation tradeoff that exists in all Bayesian search procedures[37]. Potential mitigation steps could be to sample more initially (say 5% points) or to deliberately introduce diversity into each batch of measurement points.

So far, we have reduced the diffraction image measured at each point to one single quantity (integrated intensity) in order to guide the automated experiment. These images often need to be reprocessed after the experiment to extract additional physically relevant results. Notably, the intensity distribution in the diffraction patterns contains information about the strain as well as the rotation of the crystal lattice and, in this case, the curvature of the 2D materials due to the bubbles underneath. A simple center of mass calculation in the X direction (CoMx) would yield the magnitude of the film curved in the XZ plane. The curvature (deviation of the CoMx from its nominal value) is the smallest around the center of the bubble and the largest at the edge. It also changes sign going from the left side to the right side. Center of mass calculation in the Y direction yields the magnitude of the film curved in the YZ plane. The results look slightly different from the CoMx calculations due to the way the shifted Bragg peak intersects with the Ewald's sphere. Figure 5A and B show, respectively, the CoMx and CoMy obtained from raster scan with 100% coverage on the area of interest. The unit is the number of pixel shift, relative to the center of the nominal diffraction pattern. Figure 5C and B show, respectively, the CoMx and CoMy obtained with FAST. The curvature information of the film was faithfully reproduced despite scanning just 20% of the entire area. For more information on the reconstruction of the CoM maps, the reader is referred to the "Methods" section.

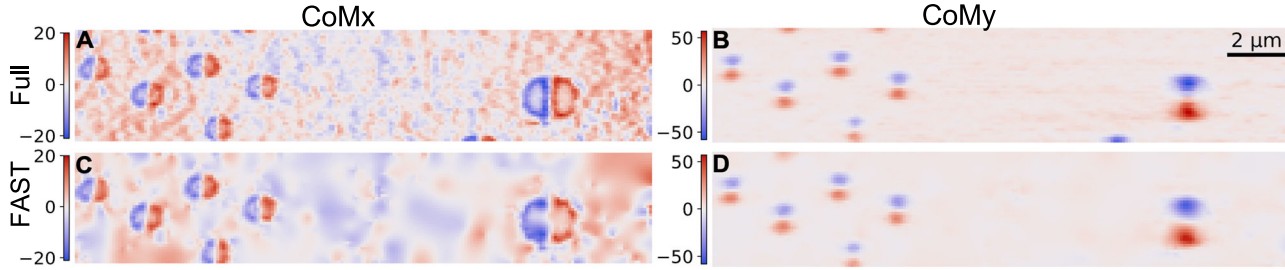

**Fig. 5 | Comparison of the per measured point center of mass (COM) of the diffraction patterns between the FAST scan at 20% coverage and full-grid scan.** Subplots **A** and **B** show the inpainted COMx and COMy, respectively, for the full-grid raster scan, and subplots **C** and **D** for the FAST scan.

## Discussion

In this work, we have showcased the FAST workflow that combines a sparse sampling algorithm with route planning to drive a scanning diffraction microscopy experiment at a synchrotron beamline. In addition to being an effective alternative to a full pointwise scan to acquire a dark-field image of the sample, FAST also produces accurate quantitative measurements of its physical properties. For our live demonstration of $200 \times 40$ points with a measurement time of 0.5 s/point, the FAST decision-making time was negligible, leading to an overall saving of $\approx 80$ min ($\approx 65\%$) of the experiment time. This saving was facilitated by our choice to acquire a batch of 50 measurements between the selection of the prospective measurement points. This ensured that the communication time stayed negligible, with no noticeable loss in the quality of points acquired when compared to a pointwise candidate selection scheme (see Supplementary Fig. S.1).

The generalizability of the FAST method comes from the fact that the key NN-based component of this workflow is trained on just the standard cameraman image[34], not on close analogs of a sample of interest. While this generalizability results in a slight loss of performance of the technique, it still shows excellent sparsity performance for cases tested in previous research[29,38] and in the current work. This has the benefit that we do not need a priori knowledge of the sample. As such, while general pre-training would be difficult to satisfy for new and expensive experiments, the FAST approach can be used directly. Furthermore, the batch prediction and route optimization approach we implement can also be directly applied in any application of choice. Moreover, the experimental application of our work uses an extensible edge device and the widely used EPICS platform for hardware control, both of which can be incorporated into any instrument even with the SLADS-Net replaced by any other sampling strategies. For example, we could just replace the dark-field detection procedure described here with a fluorescence counting setup and use exactly the FAST scheme for fluorescence-based imaging of the sample. Alternatively, since all the instruments at the APS rely on EPICS controls, one can perform transmission, surface scattering, or any other 2D scanning experiment in any applicable beamline with only minor changes to the FAST routine.

The computations in the current workflow have a time complexity of $O(2N \log N + kM \log N)$, where $N$ is the number of measured points, $M$ the number of unmeasured points, and $k$ the number of nearest neighboring measurements ($k = 10$ in our case) that we use for the feature vector calculations. Here, the first term accounts for the creation of the nearest neighbor K-d tree and the second term for the nearest neighbor calculation. The remainder of the algorithm has a linear time complexity and could be performed in parallel for the unmeasured points. We expect that it is possible to reduce this complexity using an approximate nearest neighbor search method instead of the K-d tree approach. As such, a GPU-based implementation that takes advantage of the parallelization and the approximation would likely significantly reduce the computation time. This stands in stark

contrast with the time complexity of $O(N^3)$ (for $N$ measured points) for Gaussian Processes, a similarly training-free method that is widely used for autonomous experimentation. For an illustrative example, Vasudevan et al.[20] report a GP-based scanning microscopy experiment where the calculation of each set of measurement candidates takes $\approx 6$ s on an NVIDIA DGX-2 GPU for a $50 \times 50$ image; our workflow performs an equivalent calculation for a larger $200 \times 40$ image within $\approx 1.5$ s in a low-power CPU. We note, however, that GPs remain a very powerful and generalizable approach with a bevy of applications beyond only scanning microscopy. We also note that even the current FAST decision-making time of $\approx 0.15$ s is still much larger than the typical dwell times of tens of microseconds in several popular scanning microscopy techniques (like scanning fluorescence microscopy[39]). As such, the FAST code needs to be significantly accelerated via GPU-based parallelization, approximate nearest-neighbor search methods, or other techniques, to enable its application in high-speed microscopy settings—we are looking to implement these changes in the future.

Practical applications of the FAST workflow require considerations about the spatial extent, number density, and heterogeneity of the features in the sample under investigation. Our numerical experiments for these (see Supplementary Information S.3) show that the FAST workflow is most efficient for the study of isolated sparse features as long as the features are partially sampled during the initial quasi-random scan step. Isolated features that are smaller in size than the average spacing between the initial scan points are especially likely to be missed during the initial sampling and therefore not sampled until much later in the experiment. One way to resolve this challenge is to use prior knowledge (or an informed guess) about the expected dimensions of the smallest features to tailor the density of the initial scan so that it samples almost every image patch of these dimensions. We also note that the FAST scan time increases with the increase in the overall contour (or perimeter) of the features, even if the features are at the same intensity levels and occupy the same area overall (see Supplementary Information S.3.2). Additionally, while the FAST scan is not affected adversely by heterogeneity in the feature sizes, it is less effective at resolving low-contrast features in settings with contrast heterogeneity, and addressing this can require significant prior information about the experiment (see Supplementary Information S.3.3). Moreover, we observe that FAST is less effective in experiments with a highly noisy intensity data (with signal-to-noise ratio of <0.5), but shows consistent performance in all regimes with higher signal levels (see Supplementary Information S.6). A final consideration, more practical in nature, is that the scan paths require significant motor movement, often including a retracing over points already measured. As such, there could exist scenarios in which the time required for the motor movement eclipses the time required for a single measurement. We expect to address the latter challenge by explicitly including a measurement-density-based term[38] or a movement-time-based term in the candidate selection procedure[40] or by using a line-based sampling technique[41].

Despite these considerations and challenges, we believe that the proposed FAST technique has great potential. It is an ideal tool for use cases with limited sampling or dosage budgets. It can be used to isolate regions of interest in sparse settings to prepare for pointwise scanning in these regions. More generally, it can be used to guide any scanning microscopy experiment where we do not need full pointwise information. In the future, we expect to extend this method for 3D imaging, fly scans, ptychography, and other imaging applications. We expect that these developments will significantly enhance the efficacy of scanning microscopy experiments, bolstering their use for the study of dynamic physical phenomena.

## Methods
### The SLADS-Net algorithm

The SLADS-Net algorithm[29] used within the FAST workflow is an adaptation of the Supervised Learning Approach for Dynamic Sampling (SLADS) algorithm originally developed by Godaliyadda et al.[25], and the algorithms differ only in their training approaches ("Methods: Training"). To explain the SLADS algorithm, we first denote the object we want to measure as $\mathbf{A} \in \mathbb{R}^N$, where $N$ is the total number of pixels in the image. Further, we can denote the pixel at location $1 \le s \le N$ as $a_s$ so that a measurement at the location $s$ extracts the value $a_s$; each measurement is thus characterized by the pair $(s, a_s)$. After $k$ measurements, then, we get the $k \times 2$ measurement vector

$$\mathbf{Y}^k = \begin{bmatrix} s^1 & a_{s^1} \\ s^2 & a_{s^2} \\ \vdots & \\ s^k & a_{s^k} \end{bmatrix} \tag{1}$$

Using these $k$ measurements, then, we can reconstruct (e.g., via interpolation) an estimate $\hat{\mathbf{A}}^k$ of the true object $\mathbf{A}$. The difference between $\mathbf{A}$ and $\hat{\mathbf{A}}^k$ is denoted as the distortion $D(\mathbf{A}, \hat{\mathbf{A}}^k)$ and can be calculated using any chosen metric. In the current work, we define $D(\mathbf{A}, \hat{\mathbf{A}}^k)$ to be the L2 norm:

$$D(\mathbf{A}, \hat{\mathbf{A}}^k) = ||\mathbf{A} - \hat{\mathbf{A}}^k||^2.$$

Given the measurement $\mathbf{Y}^k$ and the reconstruction $\hat{\mathbf{A}}^k$, a new measurement at any location $s$ will presumably reduce the distortion in the reconstruction. We can denote this reduction in distortion (RD) as

$$R^{k,s} = D(\mathbf{A}, \hat{\mathbf{A}}^k) - D(\mathbf{A}, \hat{\mathbf{A}}^{k,s}) \tag{2}$$

where $\hat{\mathbf{A}}^{k,s}$ is the reconstruction that includes the newly added measurement at $s$. The goal of the SLADS algorithm is then to identify the pixel location that would maximize this reduction in distortion:

$$s^{k+1} = \text{argmax}_s \, R^{k,s} \tag{3}$$

Of course, since we cannot know the value of the measurement $a_s$ or the ground truth $A$, SLADS bases its selection on the conditional expectation of reduction in distortion (ERD), which is defined as:

$$\overline{R}^{k,s} = \mathbb{E}\left[R^{k,s} | \mathbf{Y}^k\right] \quad \text{so that} \quad s^{k+1} = \text{argmax}_s \, \overline{R}^{k,s}. \tag{4}$$

The algorithm assumes that we can compute the ERD at $s$ based on just the measurement state $Y_k$ as

$$\overline{R}^{k,s} = g(\mathbf{v}^{k,s}) \tag{5}$$

where $\mathbf{v}^{k,s}$ is a location-dependent feature vector calculated using the measurement state $Y_k$. The goal of the SLADS training procedure is to estimate the function $g$.

### Training

The training procedure for the SLADS/SLADS-Net algorithm is a supervised procedure in which we generate a large number of $(\mathbf{v}^{k,s}, \overline{R}^{k,s})$ pairs and use these to estimate $g$. Note that this is a pixelwise computation that is performed independently for each measurement location $s$; for each measurement $s$ we have to calculate a reconstruction $\hat{\mathbf{A}}^{k,s}$ before we can calculate the RD $R^{k,s}$. To make this computationally tractable, Godaliyadda et al.[25] use approximations that ensure that the RD of each pixel only depends on its local neighborhood. Correspondingly, instead of working with the full measurement state $\mathbf{Y}^k$, the training procedure uses carefully designed feature vectors that capture the local neighborhood of the pixel at location $s$. As shown in Fig. 2B, the feature vector for the pixel $P$ consists of six features: (1) $\nabla_x$ and $\nabla_y$ are the spatial gradients at $P$, (2) $\sigma_{1,r}$ and $\sigma_{2,r}$ measure the deviation of the estimated value for $P$ from the nearby measured values (highlighted in red), and (3) $L$ (which is the distance of $P$ from the closest measured point) and $\rho_r$ measure the density of measurements around $P$.

The original SLADS algorithm assumes that this feature vector is linearly related to the RD, and the training therefore is a linear regression procedure. The SLADS-Net adaptation first uses a radial basis function (RBF) kernelization to transform the 6-dimensional feature vector to a 50-dimensional vector, then replaces the linear predictor with a nonlinear fully connected neural network that contains 5 hidden layers with 50 nodes each. We follow the procedure from the original SLADS-Net adaptation and use the default parameters in the Scikit-learn Python library[42] for the RBF kernelization.

In this work, we train the SLADS-Net neural network on only the standard cameraman image without using any a priori information about the sample. For the training, we generate a measurement state $\mathbf{Y}^k$ by randomly choosing a fixed number of measurement locations, then calculate the feature vector $\mathbf{v}^{k,s}$ and the RD $\overline{R}^{k,s}$ for each unmeasured pixel. We generate such sets of training pairs for 10 different sample coverage percentages between 1% and 80%. This overall comprises our training dataset. We use this data to train the neural network for 100 epochs using the Adam optimizer with a learning rate of 0.001. We use this trained model for all the simulated and experimental measurements.

### Experimental measurements

At each point of the measurement, a tight region of interest (RoI) around the expected position of the thin film Bragg peak was extracted from the corresponding diffraction image. Integrated intensities of the RoI were used to guide the NN prediction. For the flat region, the integrated intensity is high, showing up as brighter contrast on the dark field image. For the deformed region, the integrated intensity is low (darker contrast on the dark field image) as the illuminated film diffraction partially exits the selected RoI (see Supplementary Fig. S.8).

For the FAST experiment, the predicted ERD and the dark-field reconstruction served as visual guides to inform when to stop the experiment. During the experiment, we noted that the ERD and the reconstruction had stabilized by ≈20% scan coverage, but we let the experiment run to ≈35% coverage to ensure that this behavior persisted (see Supplementary Fig. S.9). While we used this visual criterion for our exploratory experiment, it is straightforward to design a numerical stopping criterion based on the absolute or relative convergence of the ERD, or on the per-iteration change in the reconstructed image.

### Statistics and reproducibility

The imaged region of the sample was selected through a visual inspection of a large-field-of-view low-resolution scan of the sample. This ensured that the high-resolution scan was directed at a region with WSe$_2$ deposition. No other statistical method was used to predetermine the sample size.

Intensity data from hot pixels were excluded during the data analysis process. No other data were excluded from the analysis.

The experiments were not randomized. The investigators were not blinded to allocation during the experiment and the outcome assessment since the described workflow provided a real-time reconstruction of the sample.

## Data availability

The numerical data used for this work are publicly available at https://github.com/saugatkandel/fast_smart_scanning[43]. The raw experimental data is publicly available at https://doi.org/10.5281/zenodo.7939730[44]. The numerical simulation data used in this work were generated using images publicly available from the MIT Libraries[34], USC-SIPI Image Database[45], and the Scikit-image software package[46]. Source data are provided with this paper.

## Code availability

The FAST software and the code for the numerical simulations are publicly available at https://github.com/saugatkandel/fast_smart_scanning[43]. The code used to analyze the experimental data is available at https://doi.org/10.5281/zenodo.7942774[44].

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

## Acknowledgements

Work performed at the Center for Nanoscale Materials and Advanced Photon Source, both U.S. Department of Energy Office of Science User Facilities, was supported by the U.S. DOE, Office of Basic Energy Sciences, under Contract No. DE-AC02-06CH11357. We also acknowledge support from Argonne LDRD 2021-0090—AutoPtycho: Autonomous, Sparse-sampled Ptychographic Imaging (awarded to M.C.). We gratefully acknowledge the computing resources provided on Bebop, a high-performance computing cluster operated by the Laboratory Computing Resource Center at Argonne National Laboratory. X.L. acknowledges support from the National Science Foundation CBET Program under award no. 2025214. A.V.B. acknowledges support from the U.S. Department of Energy, Office of Science, Office of Basic Energy Sciences Data, Artificial Intelligence and Machine Learning at DOE Scientific User Facilities program under Award Number 34532.

## Author contributions

S.K., T.Z., C.P., and M.J.C. conceived and designed the study. S.K. and T.Z. developed the workflow software. T.Z., M.H. and M.J.C. designed the synchrotron experiment. S.K., assisted by T.Z., Z.D., C.P., and M.J.C., performed the numerical simulations and the data analysis. T.Z., assisted by S.K., A.V.B., M.H., A.M., and M.J.C., performed the synchrotron experiment. X.L. and X.M. prepared the $WSe_2$ sample. A.V.B. and A.M. set up the edge computing platform. All the authors discussed the results and wrote the manuscript.

## Competing interests

The authors declare no competing interests.
