## [Peer Review File · Nature Communications]

Demonstration of an AI-driven workflow for autonomous high-resolution scanning microscopyReviewer #1 (Remarks to the Author):

The authors present a Fast Autonomous Scanning Toolkit that is based on the Supervised Learning Approach for Dynamic Sampling (SLADS) technique to perform autonomous imaging experiments in a synchrotron-based scanning X-ray microscopy setup. Here, scanning X-ray diffraction was used to capture 2D diffraction patterns in each scan point of the 008 Bragg peak of a WSe₂ sample. Essentially dark-field images were used in the automated workflow as an input. The tool takes initial measurement points from sparse, randomly selected locations and computes an initial estimate of the sample. Possible new candidate scan points are then obtained by predicting the expected reduction in distortion between a measurement and an estimate. The reduction in distortion is obtained through a linear regression neural network that essentially uses a location-dependent feature vector in a radial kernel around each point, which here contains information about the image spatial intensity gradients, the deviation of the estimated value of the unmeasured point and a nearby measurement and the density of measurements around an unmeasured point. The neural network here was solely trained on the cameraman image, which is not related to the measurement object.

The tool developed here was performed on-the-fly at the APS beam line with with an overhead of ~2% including the computation and scanning of new positions as well as processing of the diffraction patterns. The method was tested against other static sampling schemes on a pre-acquired image and performs better, since it is more efficiently sampling the regions where contrast changes occur. The authors have then demonstrated its application in a real experiment where it is found that at a 20% scan coverage a good representation of the deformed WSe₂ flakes is obtained from the integrated intensities of the diffraction patterns. A comparison of the center of mass of the Bragg peaks between a FAST scan and a full raster scan shows good agreement as well.

The presented workflow is an interesting contribution towards fully automated experimentation with little to no additional overhead, which can be performed on-the-fly in scanning X-ray microscopy experiments. The FAST procedure and general workflow is described well and the results nicely support the applicability of the method, including the linked video.

However, I would have a few questions that need to be addressed before the paper can be published in Nature Communications.

1) The authors used scanning dark-field X-ray microscopy images with low intensity (bubbles) and high intensity (WSe₂ flake) regions. Such an image representation may justify why training on the cameraman image works, it also contains low and high intensity regions. From the text it is implied that the training image does not have to be in close relation to the data (the sentence in line 281f should be checked). Does this imply that any image say from the USC-SIPI image database can be used for training? Or more specifically, how do the contrast differences and features in the training image influence the performance of FAST?

2) The architecture of the neural network is not well described. Was drop out used and if so how was it tested to ensure that there is no over fitting? What are the crucial hyper parameters needed to obtain good convergence? Related to this, how are the parameters of the radial kernel chosen and how do they affect the performance of the linear regression?

3) The authors themselves mention that the initial 1% random sampling is critical and may be a limitation if a feature is not sampled initially. How does this limitation behave if the pixel-to-feature size ratio changes significantly? And what is the limitation in the density of features (here dark regions) that FAST can capture, say if the bubble density is further reduced? On the other end of the spectrum, what if the fraction of the features in the image increases significantly? How would FAST then perform?

4) Related to this: random sampling can be done in many ways and just randomly selecting pixels may not be statistically meaningful? The description how the pixels are randomly drawn needs to be specified. How would true vs. pseudo-random number generation and hence pixel selection influence the results?

5) While in this kind of experimental setup, scan noise may not play an important role, it is intrinsic to scanning microscopy data. So to show that FAST is transferrable to other scanning imaging techniques, it would be helpful to demonstrate how noise in the data affects FAST? One can either think of scan noise, but also noise in the image intensities etc.

Reviewer #2 (Remarks to the Author):

Self-driving microscopic experiments are efficient to extract information in an optimized explorative data collection process. In this paper, the authors report the Fast Autonomous Scanning Toolkit (FAST) including a neural network, an optimization technique, and hardware control methods. This toolkit was tested in simulation and experiments with a scanning dark-field x-ray microscopy experiment. Because FAST identifies features of interest, it reduces the scan time, volume of data, and radiation dose. While this paper is useful, the whole workflow consists of only known elements; for example, the SLADS-Net algorithm used within the FAST is an adaptation of the Supervised Learning Approach for Dynamic Sampling (SLADS) algorithm with a small difference in the training stage. The novelty is generally modest. I would recommend that the paper be submitted to a specialty journal, instead of Nature Communications.

Reviewer #3 (Remarks to the Author):

In this manuscript, the authors incorporate a published dynamic sampling technique (SLADS-Net) into a novel workflow (FAST) for application to scanning dark-field microscopy. In developing this workflow, the authors have tackled many issues of general interest to the development of autonomous scanning microscopy, including initialization, route planning, and measurement batching. While I am not qualified to assess the immediate value of FAST for research that uses x-ray imaging, I think the manuscript provides a valuable conceptual framework for the development of high-speed autonomous microscopy techniques. In particular, the authors' focus on the computational cost of other approaches (such as Gaussian Processes) highlights a critical barrier to the broad adoption of state-of-the-art autonomous scanning techniques. The provided data and analysis support the conclusions with respect to the single example use case (x-ray imaging of a thin film). Claims that the FAST scheme can be directly applied to other scanning techniques could use additional support in the form of numerical experiments and a more quantitative exploration of how the image statistics of samples will impact performance. The manuscript should be published with minor revisions.

MAJOR FEEDBACK (should be addressed prior to publication)

It is important that the manuscript clearly acknowledge that the current FAST workflow is still extremely slow relative to what would be required for broad adoption in scanning microscopy. For example, typical dwell times in scanning fluorescence microscopy are 10s of microseconds (i.e., $\sim 10,000$ times faster than the x-ray measurements in this manuscript). Thus, the current FAST decision-making time for 50 points (0.15 seconds) would represent an overhead of $\sim 99\%$. While the authors do discuss using more powerful hardware, it is clear that considerable additional development will be required before the technique could be applied to many other forms of scanning microscopy.

MINOR FEEDBACK

1. The logic of pre-training on one image was unclear. To the extent such pre-training successfully generalizes across very diverse samples, it would suggest the network is learning something very simple about image statistics (such as the $1/f$ power spectra of natural images). If so, perhaps an analytical solution could be derived that is far faster than the ANN implementation.
2. Since only a few examples of simulated and experimental FAST are provided, it is very difficult to understand the domain of image statistics where FAST will provide an advantage. The authors allude to these issues in the discussion (the first two of 3 challenges), but a quantitative analysis would be far more useful than qualitative warnings. Numerical experiments varying the distribution of heterogeneous features and spatial frequency of features (relative to pixel size) could be used to quantitatively address these issues.
3. For this example of x-ray imaging, it appears that individual measurements have high SNR. In

other scanning microscopy approaches, dwell time (and thus SNR) is limited by nonlinear damage and points are resampled many times in order to achieve sufficient SNR. A discussion of how FAST would be expected perform under conditions of high measurement noise would be greatly appreciated.

REVIEWER COMMENTS

Reviewer #1 (Remarks to the Author):

The authors present a Fast Autonomous Scanning Toolkit that is based on the Supervised Learning Approach for Dynamic Sampling (SLADS) technique to perform autonomous imaging experiments in a synchrotron-based scanning X-ray microscopy setup. Here, scanning X-ray diffraction was used to capture 2D diffraction patterns in each scan point of the 008 Bragg peak of a WSe₂ sample. Essentially dark-field images were used in the automated workflow as an input. The tool takes initial measurement points from sparse, randomly selected locations and computes an initial estimate of the sample. Possible new candidate scan points are then obtained by predicting the expected reduction in distortion between a measurement and an estimate. The reduction in distortion is obtained through a linear regression neural network that essentially uses a location-dependent feature vector in a radial kernel around each point, which here contains information about the image spatial intensity gradients, the deviation of the estimated value of the unmeasured point and a nearby measurement and the density of measurements around an unmeasured point. The neural network here was solely trained on the cameraman image, which is not related to the measurement object.

The tool developed here was performed on-the-fly at the APS beam line with an overhead of ~2% including the computation and scanning of new positions as well as processing of the diffraction patterns. The method was tested against other static sampling schemes on a pre-acquired image and performs better, since it is more efficiently sampling the regions where contrast changes occur. The authors have then demonstrated its application in a real experiment where it is found that at a 20% scan coverage a good representation of the deformed WSe₂ flakes is obtained from the integrated intensities of the diffraction patterns. A comparison of the center of mass of the Bragg peaks between a FAST scan and a full raster scan shows good agreement as well.

The presented workflow is an interesting contribution towards fully automated experimentation with little to no additional overhead, which can be performed on-the-fly in scanning X-ray microscopy experiments. The FAST procedure and general workflow is described well and the results nicely support the applicability of the method, including the linked video.

However, I would have a few questions that need to be addressed before the paper can be published in Nature Communications.

1) The authors used scanning dark-field X-ray microscopy images with low intensity (bubbles) and high intensity (WSe₂ flake) regions. Such an image representation may justify why training on the cameraman image works, it also contains low and high intensity regions. From the text it is implied that the training image does not have to be in close relation to the data (the sentence in line 281f should be checked). Does this imply that any image say from the USC-SIPI image database can be used for training? Or more specifically, how do the contrast differences and features in the training image influence the performance of FAST?

Line 281 states that:

“The generalizability of the FAST method comes from the fact that the key NN-based component of this workflow is trained on just the standard cameraman image, not on close analogues of a sample of interest.”

This is correct. The NN used in the original submission was only trained on the cameraman image, which

is visually distinct from the dark-field image used in the numerical experiment (Section II.B) and the experimental demonstration (Section II.C).

We thank the referee for this intriguing question and have followed the reviewer’s suggestion to explore FAST’s performance when trained on other images from the USC-SIPi dataset. We train individually on 11 miscellaneous images shown below. We also train on a “combined” dataset that consists of all these 11 images, thus generating 12 trained NN models.

Figure 1: Miscellaneous training images

We used these models for the numerical simulation described in Section II.B and an additional (newly included) numerical simulation on the Shepp-Logan phantom image. In both these tests, the models were all similarly effective at the initial identification of the locations of the features (up to about 15% of the scan), then diverged in how they cover the sparse and low-contrast regions in the image. We want to highlight that all these models showed excellent performance at finding and scanning the high-contrast regions of the object. We can see this in the figures below, which show the progression in the SSIMs with scan coverage for the two numerical experiments, where the scans show similar progression up to SSIM of $\approx 90\%$.

Figure 2: FAST results for the Flakes simulation (left) and the Shepp-Logan phantom (right) for various training images

The fact that these images with a wide variety of structural features can all lead to excellent sparsity performance suggests that the network has learned something intrinsic to image structure and statistics. To analyze this, we looked at the training image intensities and the cumulative distribution functions as well as the power spectrum of the images. The figures are shown below:

Figure 3: Statistics for the various training images

We see that the different training images have very different image statistics from the test images for the numerical simulations, and so we cannot point to one particular feature as the cause for the performance of the FAST scan.

Identifying the intrinsic image features that drive the FAST performance is an interesting research question that we plan to explore further in the future.

We have added these experiments to the Supplemental Information, and referenced this in Section II.A:

Changes to manuscript: “We also note that, for all the results reported in this work, the underlying NN was trained on the standard “cameraman” image that has no relation to microscopy, and we discuss the choice of this specific training image in Section S.2”

Additions to supplemental material:

We have modified Section S.2 to include the miscellaneous training figures (Fig. S.2), and an expanded version of the above discussion. Also included are figures that show the progression in the SSIM and NRMSE in the FAST scans (Fig. S.4), measured points and reconstruction at 15% scan coverage for the phantom (Fig S.5) and the Flakes (Fig. S.6) cases, and the training and test image statistics (Fig S.7).

2) The architecture of the neural network is not well described. Was drop out used and if so how was it tested to ensure that there is no over fitting? What are the crucial hyper parameters needed to obtain good convergence? Related to this, how are the parameters of the radial kernel chosen and how do they affect the performance of the linear regression?

We thank the referee for pointing this out. The model used is the simple 5-layer fully-connected network with 50 nodes per layer (with ReLU activation) visualized in Fig 2 in the manuscript. We do not apply dropout. For the training, we use the Adam optimizer with a learning rate of $1e-3$ for 100 epochs.

For the RBF kernel, we use the default parameters from the scikit-learn package (https://scikit-learn.org/stable/modules/generated/sklearn.gaussian_process.kernels.RBF.html).

Our choices for the NN architecture and feature design follow the original work by Zhang et al in the SLADS-Net paper (Ref. 29 in the manuscript). We plan to explore the use of more sophisticated neural network architectures within the FAST framework in the future.

We have now clarified this in the main text of the manuscript.

Changes to manuscript:

Section II.A:

“

These feature vectors are used as input for a pre-trained neural network with 5 hidden layers, with 50 nodes per layer, and with the ReLU activation function.

”

And in Section IV.B:

“

We follow the procedure from the original SLADS-Net adaptation and use the default parameters in the Scikit-learn Python library for the RBF kernelization.

”

3) The authors themselves mention that the initial 1% random sampling is critical and may be a limitation if a feature is not sampled initially. How does this limitation behave if the pixel-to-feature size ratio changes significantly? And what is the limitation in the density of features (here dark regions) that FAST can capture, say if the bubble density is further reduced? On the other end of the spectrum, what if the fraction of the features in the image increases significantly? How would FAST then perform?

We thank the reviewer for this extremely insightful question. Based on the reviewer’s suggestion, we have added two sets of numerical experiments where: i) change the size of an isolated feature in a 128 x 128 image, and ii) where we change the size and number of features but ensure that the area covered by the features is largely unchanged.

i)

For the first case, we perform five numerical experiments with Shepp-Logan phantoms of size 128x128, 64x64, 32x32, 16x16, and 8x8. These experiments explore the case where the feature density and the feature-to-pixel size are significantly reduced. In other words, this is the case where the features are increasingly sparse. The ground truths and the FAST results for these experiments are shown below:

Figure 4: FAST performance by feature sizes

In the figure, (A) contains the ground truth images for the five numerical experiments in the top row, and just the Shepp-Logan phantoms, obtained by cropping the ground truth images, in the bottom row. The plots in (B) show the figures of metric, the SSIM and the NRMSE, for the progression of the FAST scans. The plots in (C) show the measured points and the reconstructions for the cropped regions indicated in (A) at 14% and 16% scan coverage. The figures (in B) show that the efficiency gain through

the FAST method increases as the sparsity increases. However, if the isolated feature has a size \leq the average spacing between the initial scanpoints, then the feature can be difficult to find. For this experiment, where we have a 128×128 image and 1% initial scan, the average spacing between the initial scanpoints is $\approx 12.8 \times 12.8$ pixels. Consequently, the 16×16 phantom is sampled by the initial scan, but the timepoint at which the FAST method identifies the 8×8 phantom is random (15% in this case). Once it identifies the 8×8 phantom, however, FAST immediately prioritizes and accurately reconstructs the phantom.

In fact, if we have advance knowledge of the expected feature size, then we can just calculate the initial scan pattern that ensures that almost every image patch of this size is sampled during the initial scan. Noting that the 8×8 phantom has an actual feature size of 7×5 pixels, and that the full image has size 128×128 pixels, we need 6% scan coverage to ensure that $> 99\%$ of image patches of size 7×5 pixels are sampled during the initial scan. Once we apply this consideration to the initial scan, the FAST method almost always retains an advantage over pseudorandom, random or raster scan grids for scans of isolated features.

The same calculation also informs us that for a 128×128 image, a 1% random scan is sufficient to sample $> 99\%$ of image patches that are $\geq 13 \times 13$ pixels in size. We have added a function to the FAST code to determine the required initial scan coverage based on the expected feature size.

ii)

For the second case, we perform four numerical experiments with different tilings of the Shepp-Logan phantom. These experiments explore the case where the area covered by the features remains similar, but the number of the features is different. A consequence of the change in the number of features is that the overall contour, or the perimeter, of the features also changes significantly between the experiments. The ground truths and the FAST results are shown below:

Figure 5: FAST results by feature densities

We see that the time required for the FAST scan increases with the increase in the contour due to the features, even if the features cover the same area overall. For example, 16 % scan is sufficient to resolve the outer boundaries in the single Shepp-Logan phantom setting (which we can see in the preceding figure, Figure 4, in this response). In the 2x2 phantom setting, however, we see that the outer boundaries are clearly resolved at around the 30% mark, which is as expected from the $\approx 2 \times$ increase in contour in between these settings.

On the one hand, these results show that FAST is most effective when the scan area consists of isolated features. On the other hand, even for the 8×8 case, with its dense distribution of features, FAST clearly prioritizes scanning and resolving the features over scanning the empty regions, so that the data collected is meaningful. If we further increase the number density of the features, then we get closer to a checkerboard pattern, in which case the FAST method would not present an advantage over a simple raster grid scan. We expect that in such a limiting case where the feature size is close to the pixel size, the advantage provided by any smart sampling approach would be limited.

Our experiments with the Shepp-Logan phantom also reveal that FAST with the cameraman training model struggles to resolve low-contrast features (such as the mouth of the phantom) when there are higher contrast features present in the same image. On the other hand, the numerical and experimental demonstrations included in the manuscript (Figures 3 and 4) both contain features with heterogeneous sizes but with similar contrast, and these features are all well-resolved. This suggests that the FAST method is most useful in settings where all the features, irrespective of their sizes, are at similar levels of

contrast. It is possible to address this by preprocessing the intensity input to FAST, such as to increase the contrast levels for specific intensity range of interest, but this requires significant prior information about the experiment. We hope to resolve this limitation through future work research in the SLADS-NET architecture and feature design.

We again wish to thank the reviewer for these excellent suggestions. We have added the above numerical experiments in the Supplemental Material and modified the fourth paragraph in Section III to include this discussion. The discussion now reads:

Changes to manuscript:

“

Practical applications of the FAST workflow require considerations about the spatial extent, number density, and heterogeneity of the features in the sample under investigation. Our numerical experiments for these (see Section S.3) show that the FAST workflow is most efficient for the study of isolated sparse features, as long as the features are partially sampled during the initial pseudorandom scan step. Isolated features that are smaller in size than the average spacing between the initial scan points are especially likely to be missed during the initial sampling, and therefore not sampled until much later in the experiment. One way to resolve this challenge is to use prior knowledge (or an informed guess) about the expected dimensions of the smallest features to tailor the density of the initial scan so that it samples almost every image patch of these dimensions. We also note that the FAST scan time increases with the increase in the overall contour (or perimeter) of the features, even if the features are at the same intensity levels and occupy the same area overall (see Section S.3.2). Additionally, while the FAST scan is not affected adversely by heterogeneity in the feature sizes, it is less effective at resolving low-contrast features in settings with contrast heterogeneity, and resolving this can require significant prior information about the experiment (see Section S.2.3). Moreover, we observe that FAST is less effective in experiments with highly noisy intensity data (with signal-to-noise ratio of <0.5), but shows consistent performance in all regimes with higher signal levels (see Section S.5). A final consideration, more practical in nature, is that the scan paths require significant motor movement, often including a retracing over points already measured. As such, there could exist scenarios in which the time required for the motor movement eclipses the time required for a single measurement. We expect to address the latter challenge by explicitly including a measurement-density-based term or a movement-time-based term in the candidate selection procedure, or by using a line-based sampling technique.

”

Additions to supplemental material:

We have included the above discussion and figures in Section S.3, with Section S.3.1 discussing the effect of feature size, Section S.3.2 the effect of feature density, and Section S.3.3 the effect of feature heterogeneity.

4) Related to this: random sampling can be done in many ways and just randomly selecting pixels may not be statistically meaningful? The description how the pixels are randomly drawn needs to be specified. How would true vs. pseudo-random number generation and hence pixel selection influence the results?

-

We use the low discrepancy quasi-random (LDR) Hammersly scan pattern for the initial scan. This encourages evenly spaced sampling and provides theoretical guarantees on the spacing and the

randomness of the pattern [1], while avoiding any artifacts due to a raster grid. Such LDR patterns (such as Hammersley, Halton, or Sobol patterns) are widely used for sampling experiments. In accordance with the previous literature on this, we also observe (in the numerical experiment discussed in Section II.B) that the LDR performs better than a uniform random pattern.

To avoid any confusion, we have changed in-text references to the sampling scheme from “random” to “quasi-random”.

[1] <https://www.tandfonline.com/doi/abs/10.1080/10867651.1997.10487471>

5) While in this kind of experimental setup, scan noise may not play an important role, it is intrinsic to scanning microscopy data. So to show that FAST is transferrable to other scanning imaging techniques, it would be helpful to demonstrate how noise in the data affects FAST? One can either think of scan noise, but also noise in the image intensities etc.

Following the reviewer’s suggestion, we have used the experimental data obtained in the full raster scan dark-field microscopy experiment (discussed in Section II.C in the manuscript) for a numerical simulation where we study the effect of noise in the diffraction pattern. For this, we scale the experimentally collected diffraction intensities by factors of 10, 100, and 1000, and apply Poisson noise to the scaled values, thereby simulating new intensity data with SNRs of 2, 0.62, and 0.17. Example intensity patterns for each of these cases are shown below:

Figure 6: Diffraction patterns by SNR levels

We integrate the diffraction datasets using the usual procedure (discussed in the Methods section) to obtain the ground truth microscopy patterns.

In our numerical experiments, FAST shows excellent performance even for low SNR up to the case with the SNR of 0.17, which corresponds to a maximum incident intensity of ≈ 0.24 photons (before adding shot noise) at the detector plane. We note that sparse measurement methods are generally ineffective for measurements of data with such high levels of point-to-point fluctuations. The ground truth images and the FAST simulation results at 5%, 15%, and 20% scan coverages are shown below:

Figure 7: Ground truths and FAST results by SNR

We have included these results in the supplementary material and referenced it in the main text of the paper in Section III:

Changes to manuscript: “Moreover, we observe that FAST is less effective in experiments with a high level of noise in the intensity data (with signal-to-noise ratio of ≤ 0.5), but shows consistent performance in all regimes with higher signal levels (see Section S.6).”

Additions to supplemental material:

Section S.6 includes the above discussion and figures, and the definition of the SNR we use in this discussion (Equation S.1).

Overall comment:

We want to thank the reviewer for the many excellent suggestions which, we think, have substantially improved the quality of the manuscript.

Reviewer #2 (Remarks to the Author):

Self-driving microscopic experiments are efficient to extract information in an optimized explorative

data collection process. In this paper, the authors report the Fast Autonomous Scanning Toolkit (FAST) including a neural network, an optimization technique, and hardware control methods. This toolkit was tested in simulation and experiments with a scanning dark-field x-ray microscopy experiment. Because FAST identifies features of interest, it reduces the scan time, volume of data, and radiation dose. While this paper is useful, the whole workflow consists of only known elements; for example, the SLADS-Net algorithm used within the FAST is an adaptation of the Supervised Learning Approach for Dynamic Sampling (SLADS) algorithm with a small difference in the training stage. The novelty is generally modest. I would recommend that the paper be submitted to a specialty journal, instead of Nature Communications.

Following the reviewer's comment, we have substantially increased the numerical experiments and analyses in the revised manuscript. The new analyses provide significant clarity on the domain of applications and generalizability of the proposed FAST method. We believe that the new content significantly strengthens the manuscript.

Reviewer #3 (Remarks to the Author):

In this manuscript, the authors incorporate a published dynamic sampling technique (SLADS-Net) into a novel workflow (FAST) for application to scanning dark-field microscopy. In developing this workflow, the authors have tackled many issues of general interest to the development of autonomous scanning microscopy, including initialization, route planning, and measurement batching. While I am not qualified to assess the immediate value of FAST for research that uses x-ray imaging, I think the manuscript provides a valuable conceptual framework for the development of high-speed autonomous microscopy techniques. In particular, the authors' focus on the computational cost of other approaches (such as Gaussian Processes) highlights a critical barrier to the broad adoption of state-of-the-art autonomous scanning techniques. The provided data and analysis support the conclusions with respect to the single example use case (x-ray imaging of a thin film). Claims that the FAST scheme can be directly applied to other scanning techniques could use additional support in the form of numerical experiments and a more quantitative exploration of how the image statistics of samples will impact performance. The manuscript should be published with minor revisions.

MAJOR FEEDBACK (should be addressed prior to publication)

It is important that the manuscript clearly acknowledge that the current FAST workflow is still extremely slow relative to what would be required for broad adoption in scanning microscopy. For example, typical dwell times in scanning fluorescence microscopy are 10s of microseconds (i.e., $\sim 10,000$ times faster than the x-ray measurements in this manuscript). Thus, the current FAST decision-making time for 50 points (0.15 seconds) would represent an overhead of $\sim 99\%$. While the authors do discuss using more powerful hardware, it is clear that considerable additional development will be required before the technique could be applied to many other forms of scanning microscopy.

We want to thank the reviewer for this important observation. We have added an explicit clarification in our discussion of the computational cost of the FAST method. The clarification reads (in Section 3, paragraph 3):

Changes to manuscript:

"We also note that even the current FAST decision-making time of ≈ 0.15 s is still much larger than the typical dwell times of tens of microseconds in several popular scanning microscopy techniques (like scanning fluorescence microscopy [40]). As such, the FAST code needs to be significantly accelerated via

GPU-based parallelization, approximate nearest-neighbor search methods, or other techniques, to enable its application in high-speed microscopy settings — we are looking to implement these changes in the future.”

[40] <https://iopscience.iop.org/article/10.1088/1742-6596/499/1/012002>

MINOR FEEDBACK

1. The logic of pre-training on one image was unclear. To the extent such pre-training successfully generalizes across very diverse samples, it would suggest the network is learning something very simple about image statistics (such as the $1/f$ power spectra of natural images). If so, perhaps an analytical solution could be derived that is far faster than the ANN implementation.

We thank the referee for this insightful question. We have added numerical experiments with different training images. The image descriptors we have looked at so far – the image power spectrum and contrast – are not sufficient to explain this performance. This is a curious research problem that we plan to explore further in the future.

The revised manuscript contains an exploration of FAST’s performance when trained on other images from the USC-SIPI dataset. We train individually on 11 miscellaneous images shown below. We also train on a “combined” dataset that consists of all these 11 images, thus generating 12 trained NN models.

Figure 8: Miscellaneous training images

We used these models for the numerical simulation described in Section II.B and an additional (newly included) numerical simulation on the Shepp-Logan phantom image. In both these tests, the models were all similarly effective at the initial identification of the locations of the features (up to about 15% of the scan), then diverged in how they cover the sparse and low-contrast regions in the image. We want to highlight that all these models showed excellent performance at finding and scanning the high-

contrast regions of the object. We can see this in the figures below, which show the progression in the SSIMs with scan coverage for the two numerical experiments, where the scans show similar progression up to SSIM of $\approx 90\%$.

Figure 9: FAST results for the Flakes simulation (left) and the Shepp-Logan phantom (right) for various training images

The fact that these images with a wide variety of structural features can all lead to excellent sparsity performance suggests that the network has learned something intrinsic to image structure and statistics. To analyze this, we looked at the training image intensities and the cumulative distribution functions as well as the power spectrum of the images. The figures are shown below:

Figure 10: Statistics for the various training images

We see that the different training images have very different image statistics from the test images for the numerical simulations, and so we cannot point to one particular feature as the cause for the performance of the FAST scan.

Identifying the intrinsic image features that drive the FAST performance is an interesting research question that we plan to explore further in the future.

We have added these experiments to the Supplemental Information, and referenced this in Section II.A:

Changes to manuscript: “We also note that, for all the results reported in this work, the underlying NN was trained on the standard “cameraman” image that has no relation to microscopy, and we discuss the choice of this specific training image in Section S.2”

Additions to supplemental material:

We have modified Section S.2 to include the miscellaneous training figures (Fig. S.2), and an expanded version of the above discussion. Also included are figures that show the progression in the SSIM and NRMSE in the FAST scans (Fig. S.4), measured points and reconstruction at 15% scan coverage for the phantom (Fig S.5) and the Flakes (Fig. S.6) cases, and the training and test image statistics (Fig S.7).

2. Since only a few examples of simulated and experimental FAST are provided, it is very difficult to understand the domain of image statistics where FAST will provide an advantage. The authors allude to

these issues in the discussion (the first two of 3 challenges), but a quantitative analysis would be far more useful than qualitative warnings. Numerical experiments varying the distribution of heterogeneous features and spatial frequency of features (relative to pixel size) could be used to quantitatively address these issues.

We have performed new numerical experiments to explore the effect of feature size, frequency, and heterogeneity. Specifically, we have added two sets of numerical experiments where: i) change the size of an isolated feature in a 128 x 128 image, and ii) where we change the size and number of features but ensure that the area covered by the features is largely unchanged.

i)

For the first case, we perform five numerical experiments with Shepp-Logan phantoms of size 128x128, 64x64, 32x32, 16x16, and 8x8. These experiments explore the case where the feature density and the feature-to-pixel size are significantly reduced. In other words, this is the case where the features are increasingly sparse. The ground truths and the FAST results for these experiments are shown below:

Figure 11: FAST performance by feature sizes

In the figure, (A) contains the ground truth images for the five numerical experiments in the top row, and just the Shepp-Logan phantoms, obtained by cropping the ground truth images, in the bottom row. The plots in (B) show the figures of metric, the SSIM and the NRMSE, for the progression of the FAST scans. The plots in (C) show the measured points and the reconstructions for the cropped regions indicated in (A) at 14% and 16% scan coverage. The figures (in B) show that the efficiency gain through the FAST method increases as the sparsity increases. However, if the isolated feature has a size \leq the average spacing between the initial scanpoints, then the feature can be difficult to find. For this experiment, where we have a 128 x 128 image and 1% initial scan, the average spacing between the initial scanpoints is $\approx 12.8 \times 12.8$ pixels. Consequently, the 16 x 16 phantom is sampled by the initial scan, but the timepoint at which the FAST method identifies the 8 x 8 phantom is random (15% in this case). Once it identifies the 8 x 8 phantom, however, FAST immediately prioritizes and accurately reconstructs the phantom.

In fact, if we have advance knowledge of the expected feature size, then we can just calculate the initial scan pattern that ensures that almost every image patch of this size is sampled during the initial scan. Noting that the 8×8 phantom has an actual feature size of 7×5 pixels, and that the full image has size 128×128 pixels, we need 6% scan coverage to ensure that $> 99\%$ of image patches of size 7×5 pixels are sampled during the initial scan. Once we apply this consideration to the initial scan, the FAST method almost always retains an advantage over pseudorandom, random or raster scan grids for scans of isolated features.

The same calculation also informs us that for a 128×128 image, a 1% random scan is sufficient to sample $> 99\%$ of image patches that are $\geq 13 \times 13$ pixels in size. We have added a function to the FAST code to determine the required initial scan coverage based on the expected feature size.

ii)

For the second case, we perform four numerical experiments with different tilings of the Shepp-Logan phantom. These experiments explore the case where the area covered by the features remains similar, but the number of the features is different. A consequence of the change in the number of features is that the overall contour, or the perimeter, of the features also changes significantly between the experiments. The ground truths and the FAST results are shown below:

Figure 12: FAST results by feature densities

We see that the time required for the FAST scan increases with the increase in the contour due to the features, even if the features cover the same area overall. For example, 16 % scan is sufficient to resolve the outer boundaries in the single Shepp-Logan phantom setting (which we can see in the preceding figure, Figure 4, in this response). In the 2x2 phantom setting, however, we see that the outer boundaries are clearly resolved at around the 30% mark, which is as expected from the $\approx 2 \times$ increase in contour in between these settings.

On the one hand, these results show that FAST is most effective when the scan area consists of isolated features. On the other hand, even for the 8×8 case, with its dense distribution of features, FAST clearly prioritizes scanning and resolving the features over scanning the empty regions, so that the data collected is meaningful. If we further increase the number density of the features, then we get closer to a checkerboard pattern, in which case the FAST method would not present an advantage over a simple raster grid scan. We expect that in such a limiting case where the feature size is close to the pixel size, the advantage provided by any smart sampling approach would be limited.

Our experiments with the Shepp-Logan phantom also reveal that FAST with the cameraman training model struggles to resolve low-contrast features (such as the mouth of the phantom) when there are higher contrast features present in the same image. On the other hand, the numerical and experimental demonstrations included in the manuscript (Figures 3 and 4) both contain features with heterogeneous sizes but with similar contrast, and these features are all well-resolved. This suggests that the FAST method is most useful in settings where all the features, irrespective of their sizes, are at similar levels of contrast. It is possible to address this by preprocessing the intensity input to FAST, such as to increase the contrast levels for specific intensity range of interest, but this requires significant prior information about the experiment. We hope to resolve this limitation through future work research in the SLADS-NET architecture and feature design.

We again wish to thank the reviewer for these excellent suggestions. We have added the above numerical experiments in the Supplemental Material and modified the fourth paragraph in Section III to include this discussion. The discussion now reads:

Changes to manuscript:

“

Practical applications of the FAST workflow require considerations about the spatial extent, number density, and heterogeneity of the features in the sample under investigation. Our numerical experiments for these (see Section S.3) show that the FAST workflow is most efficient for the study of isolated sparse features, as long as the features are partially sampled during the initial pseudorandom scan step. Isolated features that are smaller in size than the average spacing between the initial scan points are especially likely to be missed during the initial sampling, and therefore not sampled until much later in the experiment. One way to resolve this challenge is to use prior knowledge (or an informed guess) about the expected dimensions of the smallest features to tailor the density of the initial scan so that it samples almost every image patch of these dimensions. We also note that the FAST scan time increases with the increase in the overall contour (or perimeter) of the features, even if the features are at the same intensity levels and occupy the same area overall (see Section S.3.2). Additionally, while the FAST scan is not affected adversely by heterogeneity in the feature sizes, it is less effective at resolving low-contrast features in settings with contrast heterogeneity, and resolving this can require significant prior information about the experiment (see Section S.2.3). Moreover, we observe that FAST is less effective in experiments with highly noisy intensity data (with signal-to-noise ratio of <0.5), but shows consistent performance in all regimes with higher signal levels (see Section

S.5). A final consideration, more practical in nature, is that the scan paths require significant motor movement, often including a retracing over points already measured. As such, there could exist scenarios in which the time required for the motor movement eclipses the time required for a single measurement. We expect to address the latter challenge by explicitly including a measurement-density-based term or a movement-time-based term in the candidate selection procedure, or by using a line-based sampling technique.

”

Additions to supplemental material:

We have included the above discussion and figures in Section S.3, with Section S.3.1 discussing the effect of feature size, Section S.3.2 the effect of feature density, and Section S.3.3 the effect of feature heterogeneity.

3. For this example of x-ray imaging, it appears that individual measurements have high SNR. In other scanning microscopy approaches, dwell time (and thus SNR) is limited by nonlinear damage and points are resampled many times in order to achieve sufficient SNR. A discussion of how FAST would be expected perform under conditions of high measurement noise would be greatly appreciated.

We have used the experimental data obtained in the full raster scan dark-field microscopy experiment (discussed in Section II.C in the manuscript) for a numerical simulation where we study the effect of noise in the diffraction pattern. For this, we scale the experimentally collected diffraction intensities by factors of 10, 100, and 1000, and apply Poisson noise to the scaled values, thereby simulating new intensity data with SNRs of 2, 0.62, and 0.17. Example intensity patterns for each of these cases are shown below:

Figure 13: Diffraction patterns by SNR levels

We integrate the diffraction datasets using the usual procedure (discussed in the Methods section) to obtain the ground truth microscopy patterns.

In our numerical experiments, FAST shows excellent performance even for low SNR up to the case with the SNR of 0.17, which corresponds to a maximum incident intensity of ≈ 0.24 photons (before adding shot noise) at the detector plane. We note that sparse measurement methods are generally ineffective for measurements of data with such high levels of point-to-point fluctuations. The ground truth images and the FAST simulation results at 5%, 15%, and 20% scan coverages are shown below:

Figure 14: Ground truths and FAST results by SNR

We have included these results in the supplementary material and referenced it in the main text of the paper in Section III:

Changes to manuscript: “Moreover, we observe that FAST is less effective in experiments with a high level of noise in the intensity data (with signal-to-noise ratio of ≤ 0.5), but shows consistent performance in all regimes with higher signal levels (see Section S.6).”

Additions to supplemental material:

Section S.6 includes the above discussion and figures, and the definition of the SNR we use in this discussion (Equation S.1),

Overall comment:

We want to thank the reviewer for the many excellent suggestions which, we think, have substantially improved the quality of the manuscript.

Reviewer #1 (Remarks to the Author):

The authors have carefully addressed the referee's comments and I suggest publication of the manuscript in its present form without any further revisions needed.

The additions made by the authors to the supplemental information and the main manuscript substantially strengthen the work by highlighting the strong and weak points of FAST. The workflow will be of great interest not only to the X-ray imaging community, but makes a strong contribution to the field of autonomous experimentation.

Reviewer #3 (Remarks to the Author):

The new manuscript clearly addresses all major and minor issues raised in my initial review. I recommend publication and see no need for further revision.